# The Role of Mitochondria in Obstructive Sleep Apnea: Implications for the Upper Airway Muscles

**DOI:** 10.3390/ijms262110562

**Published:** 2025-10-30

**Authors:** Karla Carlos, Gilmar Fernandes do Prado, Celia Harumi Tengan

**Affiliations:** Department of Neurology & Neurosurgery, Escola Paulista de Medicina, Universidade Federal de São Paulo, São Paulo 04039-032, Brazil; ka.carlos1@hotmail.com (K.C.); gilmar.prado@unifesp.br (G.F.d.P.)

**Keywords:** sleep, sleep obstructive apnea, mitochondria, reactive oxygen species

## Abstract

Obstructive sleep apnea is a common but underdiagnosed sleep-related breathing disorder characterized by recurrent episodes of upper airway obstruction during sleep, leading to intermittent episodes of hypoxia and systemic consequences. Anatomical and ventilatory control factors are well-established contributors, but less is known about how mitochondria influence upper airway muscle function in this condition. As central regulators of muscle performance and cellular adaptation to hypoxia, mitochondria are particularly vulnerable to dysfunction under chronic intermittent hypoxia. Mitochondrial dysfunction increases production of reactive oxygen species, predisposing to oxidative stress, that further impairs mitochondrial function. This review focuses on the mitochondrial involvement in obstructive sleep apnea, specifically synthesizing findings on the impact on upper airway muscles. The role of mitochondrial alterations in muscle dysfunction in this context is not well understood. A better understanding of oxidative damage in these muscles may also contribute to the development of therapeutic approaches, including antioxidant strategies, to mitigate the effects of chronic intermittent hypoxia in the upper airway muscles.

## 1. Introduction

Obstructive sleep apnea (OSA) is a well-recognized but underdiagnosed sleep-related breathing disorder, characterized by recurrent episodes of upper airway obstruction during sleep [1]. Chronic intermittent hypoxia (CIH) is the main consequence and triggers a cascade of physiological and molecular responses due to transient events of oxygen desaturation followed by reoxygenation [2,3,4,5,6,7]. CIH induces adaptation responses and other effects that contribute to the development of comorbidities, such as cardiovascular disease, metabolic syndrome, and neurocognitive impairments [8].

The pathophysiology of OSA involves several factors, including anatomical features, neuromuscular function and ventilatory control mechanisms that compromise upper airway stability during sleep. Under normal conditions, during inspiration, the negative intrathoracic pressure tends to narrow the upper airway, which is kept open by the tonic activity of the upper airway muscles [9]. However, the tone of these muscles decreases significantly during sleep, increasing the risk of airway collapse. Therefore, the function of the upper airway muscles plays a crucial role in the pathogenesis of OSA.

Mitochondria are the primary source of cellular energy and important regulators of redox balance. They are central to cellular adaptations to hypoxia but are also particularly susceptible to hypoxia-induced dysfunction. In OSA, repeated cycles of CIH place particular stress on the upper airway muscles, which depend heavily on efficient mitochondrial function to sustain activity. Under hypoxia, mitochondria can have alterations in the electron transport chain, as well as in their structure and signaling pathways. Consequently, the generation of reactive oxygen species (ROS) can increase and compromise the function of these muscles.

This review was narrative in scope and focused on the mitochondrial involvement in OSA, specifically synthesizing findings on the impact on upper airway muscles. Although systemic effects on other muscle groups may also occur in OSA, we targeted upper airway muscles because of their central role in maintaining airway patency and their direct exposure to recurrent hypoxia.

To frame our discussion, we first provided a concise background on OSA, upper airway muscle function, and the role of hypoxia in mitochondrial function, summarizing the general concepts necessary for the understanding of the focus of our review.

## 2. Literature Search and Study Selection

A PubMed search was conducted to identify studies investigating mitochondrial aspects in the upper airway muscles of patients with OSA or experimental models.

The inclusion criteria were: (a) studies examining the upper airway muscles or comparable muscle cells with results related to muscle and mitochondrial structure, mitochondrial function, mitochondrial quality control or oxidative stress in OSA patients or OSA-related experimental models, (b) original research articles, and (c) written in English. Reviews, case reports, and studies not directly related to the thematic scope of this review were excluded. The search had no date restriction but was last updated on 13 October 2025.

We examined studies that assessed mitochondrial structure, function, or regulatory pathways (including biogenesis and mitophagy), along with morphological and functional features of upper airway muscles, when related to mitochondrial alterations or oxidative stress.

Due to the limited number of studies focusing specifically on oxidative stress in human upper airway muscles, we also included clinical studies evaluating systemic oxidative stress markers (in blood or exhaled breath) to provide broader insight into oxidative status in patients. The selected studies for this section met all the following criteria: (1) quantification of specific molecular oxidative biomarkers; (2) assessment of the association with OSA severity or its therapeutic modulation; (3) measurement performed in blood or other non-invasive biological fluids; and (4) primary biochemical investigation of oxidative stress.

The search strategy was based on four categories considering the topics of our review: (1) muscle and mitochondrial structure, (2) mitochondrial function, (3) mitochondrial quality control, and (4) oxidative stress. For each category, we elaborated search strings with multiple blocks that combined disease-related, anatomical, muscle-related, mechanistic, and molecular terms, as well as species-specific identifiers. Boolean operators (AND/OR) and exclusion filters (“NOT”) were applied to refine results, and additional limitations such as language and species were set using PubMed filters.

Retrieved records were evaluated considering titles, abstracts, and text content to identify studies relevant to one or more of the thematic categories. Articles that did not address these aspects or focused on unrelated tissues or outcomes were excluded. The full PubMed search strings and the number of records retrieved for each category are presented in Appendix A.

## 3. Obstructive Sleep Apnea: Clinical and Pathophysiological Aspects

OSA is defined by recurrent episodes of upper airway collapse during sleep, leading to cessation of airflow for at least 10 s despite persistent or increased respiratory effort [10,11]. The severity of OSA is assessed through polysomnography with the apnea–hypopnea index (AHI), and classified as mild (AHI ≥ 5), moderate (AHI ≥ 15), or severe (AHI ≥ 30) [12,13]. OSA is common in the adult population, particularly among individuals with obesity, advancing age, or male gender. In the general adult population, its prevalence is estimated at 9–38% for mild cases and 6–17% for moderate cases, with higher rates observed in men and a progressive increase with age [14].

Although often regarded as a disorder of mechanical upper airway obstruction, OSA is now recognized as a systemic inflammatory condition with cardiovascular, cerebrovascular, and metabolic consequences. It is associated with severe health outcomes, increased mortality risk, and high prevalence (40–60%) among individuals with cardiovascular disease [15]. The presence of common pathogenic mechanisms, such as inflammation, leads to the association of OSA with other conditions, such as psychiatric and neurological diseases [16,17,18,19,20,21].

There is a bidirectional relationship between obesity and OSA, with obesity as a major risk factor for OSA, and OSA promoting more weight gain. OSA also exacerbates the systemic inflammation already present in obesity, thereby contributing to increased disease severity. Cohort data showed that a 10% increase in body weight was associated with approximately a 30% increase in AHI [22,23,24,25].

In addition to obesity, the severity of OSA can be exacerbated by other systemic conditions, particularly chronic kidney disease and heart failure. These disorders promote rostral fluid shifts during sleep, leading to pharyngeal edema and consequent upper airway narrowing [1,26,27,28]. Furthermore, impaired clearance of inflammatory mediators in renal dysfunction may aggravate airway collapsibility. On the other hand, hypoxia secondary to OSA induces sympathetic activation, which can also impair cardiac performance in severe heart failure and contribute to disease progression [29,30,31,32].

Susceptibility to OSA is influenced by several factors, including lifestyle habits (such as smoking and alcohol consumption), individual differences in physiological responses, structural abnormalities, and age. The intensity of the responses to fluctuations in blood gases can vary among patients [11,33,34,35,36,37]. In some patients, mild hypocapnia can suppress respiratory drive, reducing pharyngeal dilator muscle activation and thereby increasing the risk of airway collapse [1]. Structural abnormalities such as a retro-positioned mandible, enlarged tongue, and excessive soft tissue, predispose the airway to increased collapsibility [27]. As OSA progresses, the airway becomes progressively more collapsible to the extent that even minor negative intraluminal pressure generated by diaphragm contraction may cause narrowing or complete collapse, leading to restricted or interrupted airflow.

To re-establish normal airflow in patients with OSA, the gold standard is continuous positive airway pressure (CPAP). CPAP is a non-invasive ventilation device that delivers positive pressure to the airway to prevent collapse [38,39]. However, the adherence to this treatment is often difficult due to discomfort, high cost, and practical difficulties in daily use. Other treatment options, such as upper airway muscle training and pharmacological interventions, including antioxidant therapies, have shown only modest benefits [40,41,42].

The development of more effective therapies depends on a deeper understanding of OSA pathophysiology, with a crucial role of the upper airway dilator muscles.

## 4. Upper Airway Muscle Function in OSA: Anatomy and Physiology

The upper airway is located in a region that lacks rigid support and is prone to collapse during sleep [9]. Thus, its patency depends on the balance between anatomical structure and the coordinated activity of several muscle groups (Figure 1A). The muscles responsible for the airway stability are located in the lateral pharyngeal walls (e.g., palatopharyngeus, superior pharyngeal constrictor), the anterior wall (e.g., genioglossus, palatoglossus), and in the soft palate (e.g., musculus uvulae). Additionally, muscles that influence hyoid position, such as the sternohyoid and geniohyoid, provide mechanical support [43]. Together, these muscles not only actively dilate and open the airway but also stiffen surrounding tissues, reducing their susceptibility to collapse under negative pressure [43].

During inspiration, a negative intrathoracic pressure draws air into the lungs, creating a pressure gradient that pulls the pharyngeal walls inward and prevents airway collapse (Figure 1B). Under physiological conditions, the collapsing force is counterbalanced by the action of pharyngeal dilator muscles, which generate forces in the anteroposterior and lateral directions to open the airway. This coordinated response is tightly regulated by respiratory centers, which send motor output primarily via the hypoglossal nerve (innervating the genioglossus) and the vagus nerve (innervating the palatoglossus, palatopharyngeus, and superior pharyngeal constrictor) [43]. However, there is a neural output that reduces muscle tone during sleep, predisposing the airway to collapse [9]. 

These muscles have a higher energetic demand in conditions with recurrent cycles of intermittent hypoxia and reoxygenation, as in OSA, which makes mitochondria a key player in upper airway muscle performance and stabilization. Hypoxia, in turn, triggers metabolic adaptations that directly engage mitochondrial pathways to preserve energy production under conditions of limited oxygen availability.

## 5. Hypoxia and Mitochondria

### 5.1. Metabolic Adaptation

Hypoxia is a state in which oxygen availability is insufficient to meet cellular metabolic demands. Hypoxia can result from several different conditions. It may occur when oxygen delivery to tissues is reduced, as in lung diseases or at high altitude, or when the blood’s oxygen-carrying capacity is diminished, as in anemia or carbon monoxide poisoning. Low intracellular oxygen can also occur when blood flow is reduced or when oxygen cannot diffuse efficiently, which is common in solid tumors. In the case of cyanide poisoning, the problem is that the cells are unable to use the oxygen that is present in sufficient levels [44].

Hypoxia triggers a wide range of adaptive responses that differ depending on the nature, duration, and severity of hypoxia, the tissue involved, the cellular context, and the underlying cause [45,46]. Although hypoxia induces physiological responses to maintain cellular homeostasis, dysregulation of hypoxia-related signaling can also contribute to the pathogenesis of several diseases, including diabetes, infections, cancer, liver disorders, cerebrovascular conditions, atherosclerosis, and neurodegenerative diseases [47].

Central to these hypoxia-driven responses are hypoxia-inducible factors (HIFs), a family of transcription factors that regulate gene expression in response to changes in oxygen availability. Among the HIFs, HIF-1 is a key isoform that acts as a master regulator of gene expression programs restoring oxygen and energy balance under hypoxic conditions [48]. HIF-1 regulates a wide range of biological processes, including metabolism, cell proliferation and survival, glycolysis, immune modulation, microbial defense, tumorigenesis, and metastasis [47].

HIF-1 is a heterodimer composed of two subunits, the α-subunit (HIF-1α) and the constitutively expressed β-subunit (HIF-1β). In normal conditions, HIF1-α is constantly degraded by the prolyl hydroxylase domain (PHD) enzyme, which requires oxygen for its activity. Thus, under normoxia, HIF-1α is kept low, whereas under hypoxia, PHD activity is suppressed, leading to HIF-1α stabilization. When stabilized, HIF-1α dimerizes with HIF-1β, forming the active form of HIF-1. HIF-1 can now initiate the expression of hypoxia-responsive genes, which will modulate processes involved in metabolic pathways, angiogenesis, erythropoiesis, cell survival, or oxygen transport [49,50,51].

HIF-1 also plays a central role in regulating mitochondrial metabolism under hypoxic conditions, modulating the tricarboxylic acid (TCA) cycle and the electron transport chain (ETC), to improve mitochondrial efficiency and reduce ROS generation (Table 1) [52].

Mitochondrial oxidative phosphorylation (OXPHOS) is the primary source of adenosine triphosphate (ATP) production in most mammalian cells and is tightly integrated with glycolysis and the TCA cycle. Under normoxic conditions, glucose is metabolized through glycolysis to produce pyruvate, which is transported into the mitochondria and converted into acetyl-CoA by the pyruvate dehydrogenase complex (PDH) [53]. Acetyl-CoA enters the TCA cycle with the reaction with oxaloacetate to form citrate via citrate synthase (CS). The cycle generates nicotinamide adenine dinucleotide (NADH) and flavine adenine dinucleotide reduced form (FADH_2_), which deliver electrons to the electron transport chain. During OXPHOS, oxygen serves as the final electron acceptor, and ATP synthase uses the resulting proton gradient to produce ATP [54]. Thus, oxygen and pyruvate are the two essential substrates for mitochondrial respiration, sustaining electron flow through the ETC and providing carbon for the TCA cycle, respectively. [53].

Under hypoxic conditions, HIF-1 induces a metabolic reprogramming that reduces mitochondrial oxygen consumption and shifts energy production away from OXPHOS. HIF-1 induces the expression of pyruvate dehydrogenase kinase (PDK), which phosphorylates and inhibits PDH, thereby blocking the conversion of pyruvate to acetyl-CoA [55]. Pyruvate is then reduced to lactate via lactate dehydrogenase A (LDHA), another HIF-1 target [56,57,58]. As a consequence, there is a decrease in pyruvate flux to the TCA cycle, thus limiting NADH and FADH_2_, and decreasing the electron flow through the ETC and oxygen consumption [55].
ijms-26-10562-t001_Table 1Table 1HIF-1–dependent targets involved in the regulation of mitochondrial function under hypoxia.HIF-1 TargetMolecular FunctionConsequence Under HypoxiaPDK (↑) [55]Inhibits pyruvate dehydrogenase, preventing conversion of pyruvate to acetyl-CoAReduces TCA cycle activity and shifts metabolism toward glycolysisLDHA (↑) [57,58]Converts pyruvate to lactate and regenerates NAD⁺ for glycolysisSupports anaerobic glycolysis and reduces mitochondrial respirationCOX4-2 (↑) [59]Alternative subunit of Complex IV with improved efficiency under low oxygenEnhances ETC efficiency during hypoxia; replaces COX4-1Lon (↑) [59]Mitochondrial protease targeting COX4-1 for degradationEnables COX4-2 incorporation into Complex IVHIGD1A (↑) [60,61]Modulates Complex IV assembly and activityIncreases Complex IV activity under hypoxiaNDUFA4L2 (↑) [62]Atypical Complex I subunit that suppresses its activityDecreases Complex I activityTMEM26B (↓) [63]Complex I assembly factorDegraded under hypoxia, reducing Complex I assembly and functionMIR210 -ISCU1/2 (↓) [64,65,66]Iron–sulfur cluster assembly proteins for ETC Complexes I–IIIDisrupts proper ETC function across multiple complexesMIR210 -NDUFA4 (↓) [67]Subunit of Complex IReduces ETC efficiencyMIR210 -SDHD (↓) [68]Subunit of Complex II (succinate dehydrogenase)Decreases Complex II activityMIR210 -COX10 (↓) [69]Complex IV assembly proteinReduction of oxygen consumptionUp or down arrows indicate increased or decreased expression, respectively. PDK: pyruvate dehydrogenase kinase, LDHA: lactate dehydrogenase A, COX: cytochrome *c* oxidase, HIGD1A: hypoxia-inducible domain family member 1A, NDUFA4L2: NADH dehydrogenase [ubiquinone] 1 alpha subcomplex subunit 4-like 2; TMEM26B: transmembrane protein 126B, MIR: microRNA, ISCU: iron-sulfur cluster assembly enzyme, NDUFA4: NADH-ubiquinone oxidoreductase, SDHD: succinate dehydrogenase complex subunit D.


### 5.2. Adaptations in Mitochondrial Quality Control

In addition to metabolic adjustments, hypoxia also triggers signaling to mitochondrial quality control pathways (Table 2) [51,70,71]. Mitochondrial quality control represents interconnected mechanisms that help to maintain function under stress [72]. Through these processes, damaged mitochondria can be removed by mitophagy, new mitochondria can be generated through mitochondrial biogenesis, mitochondrial components can be redistributed through fusion, and impaired regions can be segregated via fission [73].

Fusion and fission are dynamic processes that respond to cellular demands to preserve mitochondrial function [72]. During fusion, there is exchange of intra-organelle components, reorganization of cristae, and delay in apoptosis, thereby acting as a protective role [74]. By contrast, fission allows segregation of damaged organelles, which are then targeted for removal through mitophagy [75,76], while through mitochondrial biogenesis, these damaged mitochondria are replaced by new ones.

Several proteins are involved in regulating these processes. Fission is mainly mediated by the dynamin-related protein 1 (DRP1), which interacts with receptors such as mitochondrial fission factor (MFF), fission protein 1 (FIS1), and the mitochondrial dynamics proteins 49 and 51 (MID49, MID51) to initiate division [77,78,79,80]. The process of fusion is regulated by proteins at the outer mitochondrial membrane (mitofusin 1, MFN1, and mitofusin 2, MFN2), and at the inner membrane (optic atrophy 1, OPA1) [81,82,83]. Mitochondrial biogenesis is coordinated by transcriptional regulators, mainly peroxisome proliferative activated receptor gamma coactivator 1 α(PGC-1α), nuclear respiratory factor 1 (NRF1), and estrogen-related receptor α (ERRs), ensuring sufficient mitochondrial mass and energy supply [84].

Under hypoxia, the balance between mitochondrial fusion and fission shifts toward fission, which helps protect cells from mitochondrial damage [85]. This shift occurs because proteins that promote fission become activated, while proteins involved in fusion are reduced or inactivated [49]. In this context, during fission, DRP1 is recruited by the protein FUNDC1 rather than its usual partners MFF, FIS1, or MID49/51, promoting mitochondrial fragmentation and triggering mitophagy [86]. The expression of FUNDC1 is normally inhibited by MIR137, but under hypoxia, MIR137 levels decrease, indirectly enhancing mitophagy [87]. At the same time, hypoxia also activates the E3 ubiquitin ligase SIAH2, which promotes the degradation of A-kinase anchoring protein 121 (AKAP121), a DRP1 inhibitor, thereby increasing DRP1 activity and enhancing mitochondrial fission [71]. E3 ubiquitin ligase SIAH2 also maintains hypoxia signaling by promoting the degradation of PHD, thereby stabilizing HIF-1 [88].

Likewise, hypoxia reduces fusion through the mediators MIR17-5p and BCL2 interacting protein 3 (BNIP3). Under hypoxia, MIR17-5p levels decrease, causing a reduction in MFN2 [89] and therefore reducing fusion. In addition, the OPA1 complex is cleaved by interaction with the pro-apoptotic protein BNIP3, leading to mitochondrial fragmentation and favoring mitophagy [90,91,92]. At the same time, this effect can be counterbalanced by hypoxia-inducible domain family member 1A (HIGD-1A), which inhibits OPA1 cleavage and therefore helps preserve mitochondrial fusion [93].

In addition to classical mitophagy, it has recently been proposed that mitochondria under hypoxic conditions can be degraded via an autophagy-independent mechanism known as mitochondrial self-digestion (MSD) [94]. MSD requires both lysosomal and mitochondrial proteases and is mediated by a process called megamitochondria engulfing lysosome (MMEL). During MMEL, mature lysosomes are internalized by megamitochondria, triggering protease release, partial mitochondrial degradation, and increased mitochondrial ROS (mtROS) production [94]. MMEL is thought to depend on mitochondrial fusion, since fragmented or normally sized mitochondria appear unable to engulf intact lysosomes. It has also been proposed that megamitochondria may subsequently undergo fission to generate smaller organelles that are suitable for mitophagy [94].
ijms-26-10562-t002_Table 2Table 2Hypoxia-Induced Regulators of Mitochondrial Dynamics and Quality Control.CategoryProtein/Gene/miRNAFunctionRepercussion in HypoxiaFission machineryDRP1Mediates mitochondrial fission via recruitment to the outer membraneActivated and recruited by FUNDC1; promotes mitochondrial fragmentation and mitophagy [86]Mitophagy receptorFUNDC1Outer membrane protein that recruits DRP1 and acts as a mitophagy receptorUpregulated by hypoxia; enhances both fission and mitophagy [95,96]Fusion machineryOPA1Mediates inner membrane fusion and maintains cristae structureCleaved/disassembled by BNIP3 interaction; promotes fragmentation and mitophagy [91,92]Fusion modulatorHIGD-1AInhibits OPA1 cleavageMaintains fusion by protecting OPA1 under hypoxia [93]E3 ubiquitin ligaseSIAH2Promotes degradation of AKAP121 and prolyl hydroxylasesEnhances DRP1 activity and stabilizes HIF1, supporting adaptation to hypoxia [71]Mitophagy regulatorBNIP3Promotes mitophagy and disrupts OPA1 activityInduces mitochondrial fragmentation and facilitates mitophagy [90]Micro RNAMIR137Negatively regulates FUNDC1 expressionDownregulated under hypoxia, relieving inhibition of FUNDC1 and promoting mitophagy [87]
MIR17-5pTargets Mfn2; regulates fusion, mitochondrial integrity, and cell proliferationDownregulated in hypoxia; reduces fusion, impairs function, increases apoptosis susceptibility [89]DRP1: dynamin-related protein 1, FUNDC1: domain-containing protein 1, OPA1: optic atrophy 1, HIGD-1A: hypoxia-inducible domain family member 1A, SIAH2:seven in absentia homolog 2, BNIP3: BCL1 interacting protein 3, MIR: microRNA.


### 5.3. Reactive Oxygen Species Generation and Signaling

ROS are often considered as oxidizing agents that promote oxidative stress, but they also have a dual role, functioning as signaling molecules and participating in hypoxia-induced adaptations [97,98]. The main source of ROS generation is the ETC, but they can also be generated from several other cellular sources, including peroxisomes, lysosomes, and the endoplasmic reticulum [99]. Normally, only a small fraction of electrons escape from the ETC, but under hypoxia, ROS generation increases, particularly at Complex III [97,100]. In hypoxic conditions, Complex I undergoes a conformational change, altering the local ionic environment and reducing the fluidity of the inner mitochondrial membrane. These changes reduce the mobility of ubiquinone between Complexes II and III, thereby promoting a transient rise in ROS production at Complex III. This increase in ROS has been observed in multiple cell types, including bovine aortic endothelial cells, HeLa cells, HK-2 cells, and rat cardiomyocytes [101]. The threshold between signaling and toxicity depends on the duration and severity of hypoxia, and other factors, such as cell type, redox buffering capacity, and the source and subcellular localization of ROS. As an example, mild elevations in ROS may contribute to redox signaling processes that stabilize HIF-1 [51]. In contrast, excessive ROS can induce oxidative damage targeted to proteins, lipids, and DNA, which can contribute to mitochondrial dysfunction and cell death [98].

Increased superoxide production during hypoxia has been reported in several studies [97,100,102], although the findings are not always consistent [102]. This variability is probably due to differences in experimental conditions related to hypoxia exposure (duration, intensity, and oxygen concentration), in vitro conditions (medium composition, antioxidant content), and ROS detection method [102,103]. In humans, the influence of other diseases, genetic factors, lifestyle and dietary habits makes evaluating oxidative stress more challenging [104,105,106].

In summary, hypoxia elicits distinct mitochondrial responses, including metabolic reprogramming, alterations in quality control processes, and modulation of ROS. These changes have been documented in several tissues, particularly in the heart, brain, inflammatory cells, and tumors. However, the precise conditions under which ROS contribute to adaptive responses or cause oxidative damage remain unclear.

## 6. Mitochondrial Alterations in OSA: Influence on Upper Airway Muscles

In OSA, the upper airway muscles play an important role in maintaining the airway open, particularly during repeated cycles of hypoxia and reoxygenation. Thus, mitochondrial integrity and function are crucial for these muscles, yet this aspect has received little attention in previous reviews. To better understand muscle dysfunction and the pathophysiology of OSA, it is essential to clarify how the disease affects mitochondrial function in the upper airway muscles. In this section, we summarize the current evidence on these alterations and their functional implications.

### 6.1. Muscle and Mitochondrial Structure

Mitochondrial dysfunction can lead to alterations in mitochondrial structure and is often accompanied by changes in muscle architecture consistent with remodeling. These changes can be evaluated by histological analysis of fiber-type distribution and morphological features, as well as by ultrastructural studies. Together, these approaches provide insights into how OSA affects the structural and metabolic properties of upper airway muscles.

#### 6.1.1. Human Studies

Upper airway muscles in patients with OSA exhibited structural and functional abnormalities that may compromise their ability to maintain airway patency during sleep [107,108,109,110]. These alterations include changes in fiber type distribution and morphological features, as well as inflammatory infiltration and increased fatigability.

Human skeletal muscle fibers are classified as type I, type IIA, type IIB, and type IIX, based on ATPase activity histochemistry and myosin heavy chain isoform expression [111,112]. Type I fibers are slow-twitch and rely predominantly on oxidative metabolism, supporting endurance- and fatigue-resistant activities. Type IIA fibers are fast-twitch, fatigue-resistant, and exhibit an oxidative–glycolytic metabolism, providing both power and moderate endurance. Type IIB fibers are fast-twitch, highly fatigable, and mainly glycolytic. Type IIX fibers are fast-twitch and primarily glycolytic, specialized for short bursts of high-intensity anaerobic activity, but they fatigue rapidly. These fibers exhibit twitch properties (contraction and half-relaxation time) similar to those of type IIA and IIB fibers, with intermediate fatigue resistance [112]. In general, type II fibers have lower oxidative capacity than type I fibers, are highly energy-consuming, depend mainly on anaerobic metabolism and are appropriate for short explosive movements. While type I fibers tend to be energy-conserving, highly oxidative, and suited for prolonged low-intensity activities [113]. Therefore, analysis of fiber type distribution in muscle samples provides valuable insights into the muscle’s oxidative phenotype.

In adults with OSA, the most commonly reported alteration was an increased proportion of type II fibers, particularly type IIA, compared to controls [107,108,114,115,116,117,118,119]. This change was primarily described in upper airway muscles and was not observed in limb muscles such as the quadriceps femoris [107,109], suggesting a relationship with the specific functional demands placed on upper airway musculature.

In pediatric studies, upper airway muscles also showed a predominance of type II fibers, consistent with normal developmental patterns and not altered by OSA [120,121]. Chen et al. observed that the proportion of type I fibers decreased with age in controls and children with OSA, suggesting that fiber-type distribution may change during normal development [121]. These findings suggested that age should be considered when evaluating muscle composition in pediatric populations.

In adults, the increase in type II fibers likely represented an adaptive response to the contractile activity required to counteract airway collapsibility [115]. This interpretation was supported by the correlation between the proportion of type II fibers and disease severity [118], as well as by the reversal of fiber-type distribution toward control patterns following CPAP treatment [108]. Functionally, type II fibers produce rapid, forceful contractions needed to reopen the airway, although they are more susceptible to fatigue [122]. In fact, increased fatigability was detected by electrophysiological studies in genioglossus muscles from untreated patients [108,123]. Notably, this effect was not observed in samples from patients treated with CPAP [108], indicating that restoration of oxygen availability could reverse these changes.

Neurogenic alterations, such as angular fibers and fiber-type grouping, were also reported in the upper airway muscles of patients with OSA [117,118,124,125]. These findings suggested that a neurogenic component contributed to upper airway muscle dysfunction in OSA. One possible explanation was that repetitive snoring and apneic episodes caused progressive neuromuscular trauma, leading to sensorimotor injury that further impaired upper airway muscle function [126].

Other abnormalities in upper airway muscles varied across studies, ranging from just a type II fiber hypertrophy to myopathic features (increased connective tissue, central nuclei, and variability in fiber size) [101,108,109,110,115,118]. In addition to alterations in muscle fibers, abnormalities were reported in the intramuscular connective and vascular components, including uneven capillary distribution [110] and inflammatory infiltration [114]. The variability in these results was probably due to differences in the specific muscles examined, the histological techniques used, and the characteristics of the patient populations [127]. Comparisons across studies were limited by the fact that different muscles were sampled, including the middle pharyngeal constrictor [107,114], musculus uvulae [115,117], genioglossus [108,116], and palatopharyngeus [117,120,121]. Control samples were also heterogeneous in origin and included autopsy material from individuals who died in accidents, surgical samples from patients undergoing procedures for unrelated conditions, and samples from snorers. Furthermore, Saigusa et al. reported that the normal genioglossus showed regional variability, with a higher proportion of oxidative fibers in the posterior region, complicating the analysis [128].

Despite the variability observed at the histological level, electron microscopy studies showed alterations in the muscle structure and mitochondria (swelling, vacuolization, and reduced numbers) in the palatopharyngeus of patients with OSA [129,130]. Intra-myofibrillar lipid droplets were also observed, and their number was correlated with the apnea–hypopnea index [130], indicating a relationship with disease severity.

#### 6.1.2. Experimental Models

Experimental models clarified several effects of OSA on mitochondrial structure and function in upper airway muscles that could not be investigated in human studies. Most studies examined the genioglossus, the primary dilator responsible for maintaining airway patency, while other muscles, such as the sternohyoid and geniohyoid, were also investigated (Figure 1). The studies based on CIH models had some variations in oxygen levels, cycling conditions, and duration [131,132]. Among these, the CIH model with an N2 dilution chamber was most closely aligned with the cyclical oxygen desaturation–reoxygenation events seen in patients. In a regular protocol, rats were exposed to cycles of 2 min, including 1 min at 4–5% O_2_, followed by 1 min of reoxygenation at 21% O_2_; these cycles were repeated for 8 h daily over 5 weeks [133]. The main findings from experimental studies are summarized in Table 3.

Reported muscle abnormalities varied across studies, likely due to differences in factors such as oxygen level, duration of hypoxia, exposure time, age, and sex. Nevertheless, similar to human studies, increased proportion of type II fibers, myopathic changes and fatigability were also detected [134,135,136,137,138,139,140]. Similar results were observed in the English bulldog model, which mimics the narrowed pharyngeal airway of OSA patients [141]. In this model, the sternohyoid and geniohyoid muscles exhibited increased proportion of type II fibers and signs of myopathy (increased connective tissue, central nuclei, fiber splitting, and moth-eaten fibers) [141]. Although the experimental approaches differed, CIH models and the English bulldog model yielded findings similar to those reported in patients with OSA.

Ultrastructural studies were conducted in different experimental models, including CIH simulation in oxygen-controlled chambers (N_2_ dilution chambers) and obese animals with OSA-like features [133,138,142,143,144,145]. The *Lep^ob/ob^* obesity mouse model was included because it carries a mutation in the leptin gene that results in severe obesity and metabolic dysregulation. This model exhibits features of OSA sharing similarities with the manifestations of OSA in humans [143,146,147]. Obesity was also induced in wild-type mice by a high-fat diet (HFD), providing a model in which excess body weight develops progressively and allowing the study of obesity-related muscle alterations, including upper airway narrowing.

Across these models, ultrastructural mitochondrial abnormalities were consistently reported and characterized by swelling, reduced density, cristae disruption, and vacuolization [133,138,142,144] (Table 3). In obesity-related models, similar alterations were observed, including lipid accumulation in intermyofibrillar regions and mitochondria [138,143]. Notably, in *Lep^ob/ob^* mice, excessive intermyofibrillar mitochondria and abnormal cristae morphology were specifically reported in males [143], suggesting sex-related differences in genioglossus pathology. In agreement with this, intermittent hypoxia impaired upper airway muscle structure in male but not in female rats [148]. Although the alterations observed in obesity models could not be attributed exclusively to hypoxia, obesity is highly prevalent in patients with OSA, and such abnormalities may therefore co-exist in the upper airway muscles.

In summary, studies in humans and experimental models demonstrated that OSA was associated with structural and mitochondrial abnormalities in upper airway muscles. A higher proportion of type II fibers and fatigability were the most consistent findings in patients. However, the results varied with age, the specific muscle examined, and the characteristics of the control group. Experimental models of intermittent hypoxia and obesity reproduced many of these alterations and additionally revealed mitochondrial ultrastructural damage, including swelling, cristae disruption, and vacuolization. Together, these results suggest that both muscle composition and mitochondrial integrity were compromised in OSA, contributing to impaired muscle performance and increased airway vulnerability.
ijms-26-10562-t003_Table 3Table 3Summary of ultrastructural alterations in upper airway muscles.Specie(Muscle)ModelHypoxia Exposure ConditionsKey AlterationsHuman(palatopharyngeus)
OSAdisordered muscle fibers, decreased number of mitochondria, mitochondrial swelling, vacuolization and morphological changes [129]Human(pharyngeal constrictor)
OSADisordered myofibrils with lipid droplets, mitochondrial swelling and vacuolization [130]Rat(genioglossus)CIH model—N_2_ dilution chamber4–5% O_2_, 15–20 s duration, 30 cycles/h, 8 h/day, 5 weeksMyofibril lysis/degeneration; connective tissue proliferation; mitochondrial swelling; cristae disruption [133]Rat(genioglossus)CIH model—N_2_ dilution chamber4–5% O_2_, 15–20 s duration, 30 cycles/h, 8 h/day, 5 weeksReduced mitochondrial density; fewer subsarcolemmal and intra-myofibrillar mitochondria; swelling; reduced matrix density; disarrayed cristae [142]Rat(genioglossus)CIH model—N_2_ dilution chamber10% O_2_, 45 s duration, 20 cycles/h, 8 h/day, 4 weeksMyofibril disorganization and lysis; vacuolar degeneration; mitochondrial swelling; cristae disruption; reduced mitochondrial number and area [144]Rat(genioglossus)CIH model—N_2_ dilution chamber4–5% O_2_, 15–20 s duration, 30 cycles/h, 8 h/day, 5 weeksAutophagosomes, with cytoplasmic organelles and other vesicles encapsulated in vacuoles [145]Rat(genioglossus)Obesity mouse model *Lep^ob/ob^*-Lipid droplets; subsarcolemmal mitochondria accumulation; lipid droplet in mitochondria. In males: intermyofibrillar mitochondria accumulation; abnormal cristae morphology [143]Mouse(genioglossus)HFD-induced obese mouse-Vague myofibril structure; reduced mitochondria number; swollen mitochondria; cristae disintegration [138]Hypoxia exposure is shown as: O_2_ level during hypoxia, duration of hypoxia per cycle, number of cycles per hour, daily hours of exposure, and total regimen duration. CIH: chronic intermittent hypoxia, HFD: high fat diet.


### 6.2. Mitochondrial Function

Mitochondrial function in OSA was examined through various methodological approaches in human samples and experimental models. Together, these approaches provided an integrated perspective encompassing enzymatic activity, protein expression, and mitochondrial bioenergetics. A summary of findings from human and experimental studies is provided in Table 4.

#### 6.2.1. Human Studies

Human studies focused on upper airway muscles, such as the palatopharyngeus, the musculus uvulae, and the genioglossus. To evaluate mitochondrial function, investigators primarily used enzyme activity assays and histochemistry to assess glycolytic capacity, TCA cycle activity, and OXPHOS function.

Biopsies from the upper airway muscles of patients with OSA revealed selective metabolic changes. Increased activities of glycolytic enzymes, including phosphofructokinase (PFK), glyceraldehyde-3-phosphate dehydrogenase (GAPDH), and glycogen phosphorylase (PYG), were observed in uvula samples from patients with OSA compared with snorers [115,116]. These findings suggest an adaptive response to hypoxia, with a shift in energy metabolism toward glycolysis, which is consistent with the rapid, intermittent muscle activation required during apnea episodes [55]. However, this metabolic shift may be detrimental to upper airway muscles, which function continuously, since glycolysis provides limited ATP and promotes lactate accumulation, leading to acidification and potential muscle damage.

These alterations were not observed in the genioglossus muscle, which exhibited no major changes in glycolytic enzyme activity compared with snorers [116]. Such discrepancies may reflect the characteristics of the comparison group, as snorers are not entirely healthy controls, as well as potential differences in the metabolic susceptibility of upper airway muscles.

Glucose uptake in the tongue was also evaluated using a non-invasive in vivo approach with positron emission tomography (PET) and [18F]-2-fluoro-2-deoxy-D-glucose (FDG). Using this technique, Kim et al. observed reduced glucose uptake in the genioglossus muscle of patients with OSA. This reduction was attributed to the predominance of type II fibers in this muscle, which exhibit lower glucose transporter type 4 (GLUT-4) expression and reduced hexokinase activity [149].

Only a few studies assessed enzymes of the TCA cycle, and the findings were not entirely consistent [115,150,151]. Analyses of the musculus uvulae showed no change in CS activity.

With respect to OXPHOS, human studies have been limited to histochemical analyses of Complex IV in the musculus uvulae and palatopharyngeus, which revealed a small proportion of COX-negative fibers (1.4–3.2%) [115,152].

Taken together, although studies remain relatively scarce, the available evidence suggests that upper airway muscles in OSA undergo a metabolic shift characterized by increased glycolytic activity and, in some cases, reduced glucose uptake and altered oxidative enzyme function. However, these changes were muscle-specific, indicating that not all upper airway muscles respond similarly to the physiological demands of OSA.

#### 6.2.2. Experimental Models

In animal models, most studies have focused on the genioglossus and sternohyoid muscles exposed to CIH or a high-fat diet. The analytical methods included enzyme activity assays, gene expression analyses, Western blotting, proteomics, immunohistochemistry, and functional probes to assess mitochondrial membrane potential.

In contrast to the changes observed in the human musculus uvulae, glycolytic enzyme activities were not altered in the sternohyoid muscle of rats exposed to CIH [151,153]. Regarding the TCA cycle, reduced CS expression was reported in the genioglossus of rats subjected to CIH [142], whereas no change was observed in the sternohyoid in mice.

Complementary analyses using histochemistry, mRNA expression, and proteomics revealed that several components of the respiratory chain were affected [133,138,142,154,155]. The exception was Complex II, for which the results were inconsistent. Reduced SDHA expression was reported in the genioglossus using immunohistochemistry [154], while histochemical analysis of the sternohyoid showed no change in SDH activity [139,140,153].

Complex IV was the most consistently affected complex across studies [133,142,154,155]. A small proportion of COX deficient fibers was detected in the rat genioglossus by histochemistry [142], which aligned with reduced Complex IV activity in enzyme assays [133], decreased expression of Complex IV subunits [142], and reduced expression of a Complex IV assembly protein [155].

Additional signs of functional impairment included a reduction in mitochondrial membrane potential (ΔΨm) in the genioglossus of CIH-exposed rats [133]. This finding was supported by the decrease in adenine nucleotide translocase 1 (ANT1) expression, suggesting that ATP transport was impaired and energy homeostasis was disrupted [142].

In summary, these studies suggest that upper airway muscles in OSA exhibit metabolic alterations, including a shift toward glycolysis and impairment of ETC function. The results showed the involvement of Complexes I, III, IV, and V, while the involvement of Complex II remains uncertain.
ijms-26-10562-t004_Table 4Table 4Alterations in Mitochondrial Function in the Upper Airway Muscles from OSA Patients and Experimental Models.Pathway/Protein/EnzymeReported ChangeMuscle (Model)MethodsGlucose uptake↓Genioglossus (human)PET/FDG imaging [149]Glycolysis


HKNo change (vs. snorers)Musculus uvulae (human)Enzyme assays [115]PFKNo change (vs. snorers)Genioglossus (human)Enzyme assays [116]
↑ (vs. snorers)Musculus uvulae (human)Enzyme assays [115,116] GAPDHNo change (vs. snorers)Genioglossus (human)Enzyme assays [116]
↑ (vs. snorers)Musculus uvulae (human)Enzyme assays [115,116] 
No change Sternohyoid (rat, CIH)Histochemistry [153]
No change Sternohyoid (rat, CIH)Enzyme assays [151]PYGNo change (vs. snorers)Genioglossus (human)Enzyme assays [116]
↑ (vs. snorers)Musculus uvulae (human)Enzyme assays [116]TCA Cycle


CSNo change (vs. snorers)Musculus uvulae (human)Enzyme assays [115] 
No changeSternohyoid (mouse, CIH)Enzyme assays [150,151]
↓Genioglossus (rat, CIH)mRNA expression [142]OXPHOS


Complex I↓ ActivityGenioglossus (rat, CIH)Enzyme assays [133]
↓ NDUFC2, NDUFAB1Genioglossus (mouse, CIH)Proteomics [155]Complex II↓ SDHAGenioglossus (mouse, CIH)IHC [154]
No change in SDH activitySternohyoid (rat, CIH)Histochemistry [139,140,153]Complex III↓ UQCRC2 (Core protein 2) *Genioglossus (rat, HFD)WB [138]
↓ UQCRBGenioglossus (mouse, CIH)Proteomics [155]Complex IVCOX negative in fibers (1.4%)Musculus uvulae (human)Histochemistry [152]
COX negative in fibers (3.2%)Palatopharyngeus (human)Histochemistry [152]
No change (vs. snorers)Musculus uvulae (human)Enzyme assays [115]
↓ COX (low % fibers)Genioglossus (rat, CIH)Histochemistry [142]
↓ ActivityGenioglossus (rat, CIH)Enzyme assays [133]
↓ COX4I1Genioglossus (rat, CIH)mRNA expression [142]
↓ COX7C, COX4I2, COX6CGenioglossus (mouse, CIH)Proteomics [155]
↓ COX17 (assembly protein)Genioglossus (mouse, CIH)Proteomics [155]Complex V↓ ATP5F1A (α subunit) *Genioglossus (rat, HFD)WB [138]
↓ ATP5MF, ATP5ME, ATP5MGGenioglossus (mouse, CIH)Proteomics [155]Mitochondrial function


ΔΨm (Mitochondrial membrane potential)↓Genioglossus (rat, CIH)Fluorescent probe [133]ANT1↓Genioglossus (rat, CIH)mRNA expression [142]* protein detected by the total OXPHOS rodent WB antibody cocktail. ANT1: adenine nucleotide translocator 1; CIH: chronic intermittent hypoxia; COX: cytochrome c oxidase (Complex IV); CS: citrate synthase; ΔΨm: mitochondrial membrane potential; GAPDH: glyceraldehyde-3-phosphate dehydrogenase; HK: hexokinase; IHC: immunohistochemistry; Mfn2: mitofusin 2; NRF1: nuclear respiratory factor 1; OXPHOS: oxidative phosphorylation system; PFK: phosphofructokinase; PYG: glycogen phosphorylase; SDHA: succinate dehydrogenase subunit A; SDH: succinate dehydrogenase; UQCRC2: ubiquinol-cytochrome c reductase core protein 2; UQCRB: ubiquinol-cytochrome c reductase binding protein; PET: positron emission tomography; FDG: [18F]-2-fluoro-2-deoxy-D-glucose; WB: Western blot; ↑: increased; ↓: decreased.

### 6.3. Mitochondrial Quality Control

Mitochondrial quality control involves coordinated processes that preserve mitochondrial integrity and function. Investigation of abnormalities in quality control processes aimed to detect ultrastructural alterations in mitochondrial integrity and shape, as well as changes in mtDNA copy number and in the expression of key regulators involved in fusion, fission, biogenesis, and mitophagy. A summary of the studies addressing these processes is presented in Table 5.

#### 6.3.1. Human Studies

Only a limited number of studies have examined mitochondrial biogenesis in OSA, and most have used blood samples. These studies primarily assessed mitochondrial DNA (mtDNA) copy number, along with markers related to biogenesis signaling, but the results were inconsistent [156,157,158]. Reduced mitochondrial biogenesis was suggested by a decrease in mtDNA copy number in one study [157]. In contrast, another study reported increased mtDNA copy number together with higher PGC-1α expression, suggesting the opposite [156]. Lin et al. further showed that PGC-1α expression and mtDNA copy number were increased in blood but not in exhaled breath condensate from the same individuals, indicating that findings may depend on the type of biological sample analyzed [158].

Because systemic measurements may not accurately reflect what actually happens at the cellular level in upper airway muscles, direct analysis of muscle tissue provides a more reliable approach for assessing mitochondrial regulation. However, studies analyzing upper airway muscles directly were rare. Chen et al. examined the palatopharyngeus muscle of patients with OSA and observed decreased PGC-1α expression together with a reduced number of mitochondria on electron microscopy, indicating decreased mitochondrial biogenesis [129].

#### 6.3.2. Experimental Models

In contrast to the inconsistent findings regarding mitochondrial biogenesis in blood samples described in the previous section, studies using rodent CIH models showed reduced mitochondrial biogenesis in the genioglossus, as indicated by lower expression of key regulatory factors and decreased mitochondrial density [142]. Additionally, these results were supported by the decreased expression of PGC-1α and NRF1 in C2C12 myoblasts exposed to intermittent hypoxia [129]. Simultaneously, there were also changes in mitochondrial morphology and function, including reduced enzymatic activity of complexes II and IV [142].

In the HFD model, mitochondrial fission was increased in the genioglossus with higher DRP1 and lower Mfn2 expression, suggesting mitochondrial fragmentation [138]. While in the CIH model, mitophagy-related markers were also increased and associated with impaired mitochondrial function and altered ultrastructure [145].

In addition, several studies reported enhanced apoptosis in CIH rodent models [138,159]. Wang et al. showed that five weeks of CIH exposure led to progressive apoptosis in the genioglossus, with increased levels of cleaved caspase-3, suggesting that apoptotic processes may begin early during hypoxic stress [145]. These results were supported by findings demonstrating activation of apoptosis-related mechanisms at the gene expression level after six weeks of CIH exposure [160]. Experiments with genioglossus myocytes showed a marked increase in apoptosis after 12 weeks of CIH exposure, suggesting that prolonged hypoxia exacerbated apoptosis in a time-dependent manner [138,159]. Activation of the apoptotic pathway was observed in the HFD model, with increased DNA fragmentation, increased Bax/Bcl-2 ratio, and cytochrome c release into the cytoplasm [138,159].

Together, these alterations suggested that mitochondrial homeostasis was disrupted, with reduced biogenesis and increased fission, favoring the accumulation of fragmented mitochondria, triggering mitophagy and, when damage persisted, the activation of the apoptotic pathway.

In summary, findings from human studies were inconsistent regarding mitochondrial biogenesis due to the inadequacy of the samples analyzed. Studies conducted in experimental models revealed alterations in mitochondrial quality control, characterized by reduced biogenesis, increased fission, evidence of mitophagy, mitochondrial dysfunction, and increased apoptosis. However, given the limited number of available studies, these observations require further confirmation.

Under physiological conditions, a balance between mitophagy and biogenesis maintains energy supply and cell survival by removing damaged mitochondria and generating new ones [161]. When this balance is disrupted, either through excessive mitophagy or insufficient biogenesis, mitochondrial function declines, leading to muscle weakness and reduced ability to maintain upper airway patency. In addition, a shift toward mitochondrial fission and mitophagy is likely to increase ROS generation. In this context, examining ROS and oxidative damage is particularly relevant, as they can further exacerbate mitochondrial dysfunction.

### 6.4. Oxidative Stress

Oxidative stress reflects an imbalance between ROS production and the capacity of antioxidant defenses to neutralize them. Because species such as superoxide, nitric oxide, and peroxynitrite are highly reactive and have very short half-lives, they are rarely detected directly. Instead, oxidative stress is typically assessed by measuring oxidized targets, such as DNA, proteins, and lipids, or by quantifying antioxidant levels [99]. In OSA, oxidative stress has been investigated mainly through systemic biomarkers in patients and through direct analyses of upper airway muscles in experimental models. A summary of these studies is presented in Table 6.

#### 6.4.1. Human Studies

Studies evaluating oxidative stress in the upper airway muscles of OSA patients were very limited. Kimoff et al. examined surgical specimens of the uvula and palatal tissue (primarily the palatopharyngeus muscle) and detected increased protein carbonylation, a marker of oxidative stress, which was higher in severe compared to mild OSA cases; however, non-apneic control samples were not included [162].

Although studies investigating the tissue directly are ideal, they require invasive procedures to obtain samples, and appropriate controls are not easily available. Upper airway muscle biopsies are typically collected during surgical procedures for OSA, which may involve postoperative discomfort and potential risk to already compromised tissues. Consequently, most patient studies have relied on analyses of more accessible samples such as serum, plasma, or urine, with a focus on identifying disease-associated biomarkers.

Given our focus on upper airway muscles, we included patient-based studies analyzing blood or exhaled breath condensate to provide contextual information (see Section 2 and Appendix A for selection criteria). Systematic reviews reported a similar pattern of oxidative stress alterations in OSA, indicating that the examples presented here reflect the broader findings in the literature [163,164].

Overall, these studies pointed to increased oxidative stress in OSA (Table 6), with evidence of lipid peroxidation (↑ 8-isoprostane, malondialdehyde MDA), protein oxidation (↑ protein carbonyls), and increased ROS generation (↑ hydrogen peroxide), accompanied by reduced antioxidant defenses involving enzymatic (superoxide dismutase, SOD, catalase) and non-enzymatic components (glutathione, GSH; total antioxidant status/ total antioxidant capacity, TAS/TAC) [162,165,166,167,168,169,170,171,172]. The overall pattern supported the presence of an oxidative imbalance in OSA, characterized by increased oxidant production and impaired antioxidant responses, which was related to disease severity [168] and partially reversible after CPAP therapy, as indicated by reductions in 8-isoprostane, MDA, and hydrogen peroxide [166,167,172].

However, when analyzing systemic samples, it is important to emphasize that the choice of biological material is critical. The reliability of systemic measures in reflecting the oxidative status of upper airway muscles remains uncertain, and another challenge is that these markers are influenced by comorbidities, genetic susceptibility, lifestyle, and diet [104,105,106]. In a large cohort study a multiple regression analysis on three markers of oxidative stress (lipid peroxidation with 8-isoprostane), DNA oxidation with 8-OHdG, and SOD), Peres et al., found that only 8-isoprostane was independently associate with AHI, after adjusting for demographic, clinical, and lifestyle variables, including body mass index, AHI, comorbidities, statin use, and ethnicity [173]. This study shows the complexity of evaluating oxidative stress markers in patients and the importance of controlling for confounding factors in biomarker studies.

The limited availability of human upper airway muscle tissue underscores the importance of experimental models for studying mitochondrial and oxidative changes in OSA. These models not only provide direct access to the tissue of interest but also enable the investigation of mechanisms underlying oxidative stress and its impact on muscle function.
ijms-26-10562-t006_Table 6Table 6Studies assessing oxidative stress and antioxidant markers in patients with OSA.ProcessMarkerSampleBasalWith CPAPStudyProtein oxidationProtein carbonylBloodNo changen.d.[170]Protein carbonylationUvula, palatopharyngeusPresentn.d.[162]AOPPBlood↑No change[165]Lipid peroxidation8-IsoprostaneExhaled breath and blood↑↓[166]8-IsoprostaneBloodNo changen.d.[171]8-IsoprostaneBloodNo changen.d.[170]MDABlood↑↓[172]TBARSBloodNo changen.d.[171]TBARSBlood↑n.d.[168]TBARSBloodNo changen.d.[170]ROS productionHydrogen peroxideBlood↑↓[167]Hydrogen peroxideExhaled breath condensate (children)↑n.d.[169]Antioxidant defenseGSHBlood↑n.d.[170]GSHBlood↓
[165]CatalaseBloodNo changen.d.[170]SODBloodNo changen.d.[170]SODBlood↓n.d.[168]FRAPBlood↓↑[165]TASBlood↓n.d.[168]TACBloodNo changen.d.[170]Abbreviations: CPAP: Continuous positive airway pressure; AOPP: Advanced oxidation protein products; ROS: Reactive oxygen species; MDA: Malondialdehyde; TBARS: Thiobarbituric acid reactive substances; FRAP: ferric reducing antioxidant power; TAS: Total antioxidant status; TAC: Total antioxidant capacity; GSH: Reduced glutathione; GSSG: Oxidized glutathione; SOD: Superoxide dismutase; n.d. = not determined.↑ = increased; ↓ = reduced.

#### 6.4.2. Experimental Models

Experimental studies directly analyzing upper airway muscles can be grouped into two main categories: (i) those assessing oxidative stress markers and (ii) those testing the effects of antioxidant or pro-oxidant substances. The main findings are summarized in Table 7.

Evidence of oxidative stress was reported in rodent models and cultured myoblasts [138,145,159,160,174]. CIH models demonstrated increased ROS production, lipid peroxidation, reduced antioxidant enzyme activity and transcriptional changes related to ROS responses in the genioglossus [145,159,160,174,175], while protein oxidation was not detected in the sternohyoid muscle [140].

Across studies, most assessments focused on overall ROS levels without distinguishing their subcellular origin. Only one study specifically examined mtROS, reporting increased mtROS in isolated genioglossus mitochondria under CIH [175].

In HFD models, oxidized DNA and lipid peroxidation were also observed [138]. In the same study, C2C12 myotubes treated with palmitic acid showed increased mtROS; however, this approach modeled HFD conditions without reproducing CIH, which is a limitation.

Additional insights came from studies evaluating antioxidant and pro-oxidant interventions. In the rat sternohyoid muscle, antioxidant treatment improved electrophysiological responses [148,176,177,178], while pro-oxidant treatment with buthionine sulfoximine (BSO) exacerbated CIH-induced fatigue [176]. However, antioxidant effects differed depending on the compound. Tempol, a superoxide scavenger, enhanced muscle force and performance with acute administration [177,178] and reversed CIH-induced weakness after nine days of treatment [177]. In contrast, N-acetylcysteine (NAC), which acts via the glutathione system, did not improve muscle function in vitro [148,176]. Treatment with genistein reduced mtROS and lipid peroxidation, enhanced antioxidant enzyme activity, and increased resistance to fatigue, indicating a protective effect against oxidative damage [175].

Overall, results from patient studies and experimental models suggested that oxidative stress is a consistent feature in OSA. Systemic studies have demonstrated increased levels of oxidative stress markers; however, their interpretation remains limited by the types of samples analyzed and by confounding factors. In contrast, experimental models provided direct evidence of oxidative stress in upper airway muscles, although most studies measured overall ROS levels without distinguishing their intracellular origin. Together with the findings from antioxidant and pro-oxidant interventions, these observations suggest that oxidative stress plays a relevant role in the pathogenesis of upper airway muscle dysfunction in OSA.

The main findings related to mitochondrial alterations found in the upper airway muscles are summarized in Figure 2.

## 7. Conclusions and Future Perspectives

Studies in both humans and experimental models showed that the upper airway muscles undergo substantial alterations in response to CIH. However, most of the detailed mechanistic evidence comes from experimental models (Table 8) due to the limited availability of human muscle tissue. At first, hypoxia may induce adaptive changes to maintain muscle function under reduced oxygen availability. Muscle fibers are remodeled, energy metabolism is adjusted, and mitochondrial quality control processes are activated to help maintain sufficient ATP production. However, these compensatory responses have limited capacity, and prolonged exposure to CIH with decreased respiratory chain activity, can lead to increased ROS generation, disrupting the balance between ROS and antioxidant defenses. Oxidative damage may affect mitochondria, especially the OXPHOS system, leading to mitochondrial dysfunction and further predisposing to increased ROS production. This way, a self-perpetuating cycle of mitochondrial dysfunction and oxidative stress is established, compromising upper airway muscle function, increasing airway collapsibility, and further exacerbating hypoxic events. A similar mechanism was previously proposed, considering that adaptation to hypoxia starts with a compensatory hyperactivation of the upper airway dilator muscles, but later a structural injury is established, leading to reduced effectiveness of these muscles and hypoxia, which again promotes hyperactivation of the dilator muscles [122]. Muscles with insufficient OXPHOS are unable to maintain full function, and as oxidative lesions increase, the tendency is for a progressive decline in muscle strength and damage. Thus, what begins as an adaptive response to hypoxia may progress into a maladaptive state culminating with structural injury.

Similarly to the upper airway muscles, mitochondrial alterations have also been reported in diseases frequently associated with OSA, such as cardiovascular and metabolic disorders [179]. CIH caused ultrastructural changes in mitochondrial morphology, reduced SOD activity, increased mtROS production, and induction of apoptosis in endothelial cells [180]. Pancreatic islets exposed to CIH displayed increased mtROS and impaired insulin synthesis [181]. In the myocardium, in addition to mitochondrial ultrastructural disruption, impaired cardiac remodeling and dysfunction were observed, associated with increased mitochondrial fragmentation, reduced oxygen consumption, and increased ROS [182,183]. Treatment with mitochondrial-targeted antioxidants, such as MitoTempo or MitoTempol, reversed endothelial apoptosis and restored insulin synthesis and glucose homeostasis in pancreatic islets [180,181].

The improvement in upper airway muscle function with antioxidant treatment in experimental studies further supported the involvement of oxidative stress in muscle dysfunction. Treatment with genistein reduced genioglossus fatigue by reducing oxidative stress and increasing mitochondrial biogenesis [175]. Orientin also promoted a muscle effect by increasing fatigue resistance and the proportion of type I fibers, and increasing mitochondrial biogenesis [184]. However, genistein and orientin are not specifically targeted to mitochondria. Considering that mitochondria are the major sources of ROS, mitochondria-targeted antioxidants such as mitoquinone (MitoQ) and MitoTEMPO appear particularly promising due to their localized action within mitochondria. Although MitoQ has undergone clinical evaluation in other diseases, no preclinical studies or registered clinical trials have yet tested MitoQ or MitoTEMPO in OSA (clinicaltrials.gov, accessed on 23 October 2025).

In conclusion, CIH may initially induce mitochondrial and metabolic adaptations to preserve muscle function under hypoxic stress. However, when the adaptive threshold is exceeded, oxidative damage and muscle dysfunction can accelerate disease progression. The interplay between mtROS production, insufficient antioxidant defenses, and ongoing mechanical and hypoxic stress appears central to the pathogenesis of upper airway muscle impairment in OSA.

## Figures and Tables

**Figure 1 ijms-26-10562-f001:**
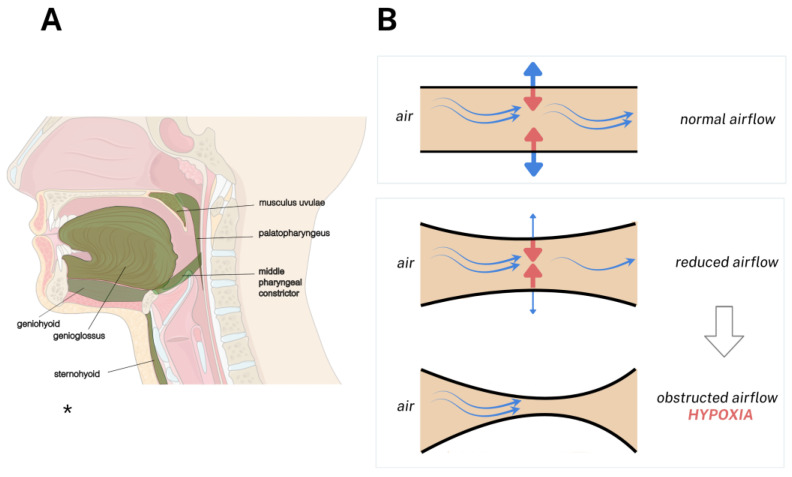
Mechanism of Upper Airway Obstruction in OSA. The airway patency is maintained by the synergistic action of upper airway muscles, which counteract the collapsing force of negative inspiratory pressure. Panel (**A**) illustrates the muscles investigated in patients and experimental models, highlighted in light green. A schematic figure of the oropharynx (**B**) is shown on the right to demonstrate the upper airway flow in OSA. The upper panel shows a normal condition in which the dilator muscle activity effectively (blue arrow) opposes the negative pressure (red arrow), keeping the airway open. The middle panel represents a partial airway collapse, where reduced dilator muscle tone fails to counterbalance the negative pressure, resulting in diminished airflow. This imbalance leads to complete airway obstruction and consequent hypoxia in OSA (lower panel). * Image on the left was adapted from Servier Medical Art (https://smart.servier.com/, accessed on 3 July 2025), licensed under CC BY 4.0 (https://creativecommons.org/licenses/by/4.0/, accessed on 3 July 2025).

**Figure 2 ijms-26-10562-f002:**
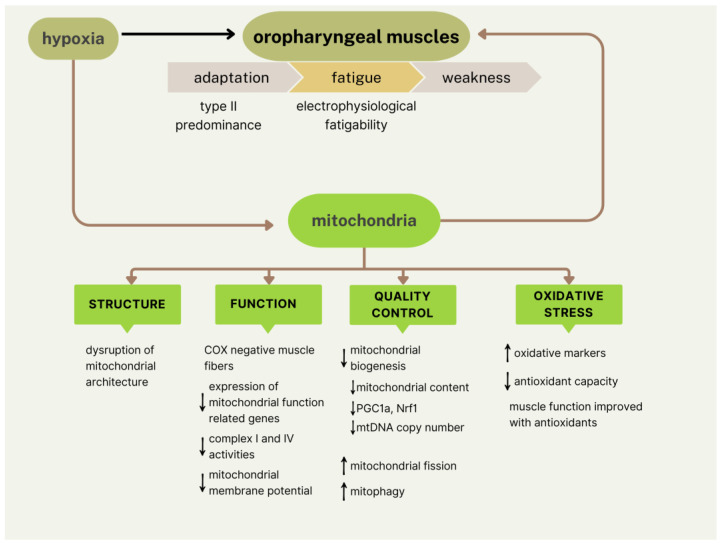
Summary of mitochondrial alterations in upper airway muscles in OSA. We illustrate the upper airway muscle abnormalities observed in patients with OSA (on top), which lead to adaptation to hypoxia, fatigue, and weakness. At the bottom, we present the hypoxia-induced mitochondrial alterations in structure, function, quality control, and markers of oxidative stress. These alterations impact muscular function. ↑ = increased; ↓ = reduced; COX: cytochrome c oxidase; PGC1a: peroxisome proliferative activated receptor gamma coactivator 1 alpha; Nrf1: nuclear respiratory factor 1; mtDNA: mitochondrial DNA.

**Table 5 ijms-26-10562-t005:** Alterations in Mitochondrial Quality Control in OSA Patients and Experimental Models.

Process	Alt.	Molecular Alterations	Mitochondrial Function	Methods	Samples (Model)
Biogenesis	↑	↑mtDNA	↑ Oxidative stress;	qPCR, oxidative stress assay (d-ROM test)	Human blood cells [156]
	↓	↓ mtDNA	n.d.	qPCR	Human whole blood DNA [157]
	↑	↑ PGC-1α ↑ mtDNA	n.d.	qPCR, ELISA	Human exhaled breath condensate [158]
	-	- PGC-1α - mtDNA	n.d.	qPCR, ELISA	Human blood [158]
	↓	↓ PGC-1α ↓ NRF-1	↓ number of mitochondria	qPCR, WB, EM	Palatopahryngeus muscle [129]
	↓	↓ PGC-1α ↓ ERRα ↓ NRF1	↓ mitochondrial density structural disruption	EM, qPCR, histochemistry	Rat genioglossus (CIH) [142]
	↓	↓ PGC-1α ↓ NRF1	n.d.	qPCR, WB	C2C12 myoblasts (intermittent hypoxia) [129]
Fission	↑	↑ DRP1 ↓ Mfn2	↓ Complex III ↓ Complex V ↓ Number of mitochondria ↓ mtDNA	WB, EM, qPCR	Rat genioglossus (HFD) [138]
Mitophagy	↑	↑ p-ULK1 ↑ p-Raptor ↑ SIRT1 ↑ TSC1 ↑ LC3-II	altered mitochondrial structure ↓ mitochondrial function	WB, qPCR, EM	Rat genioglossus (CIH) [145]
Apoptosis	↑	↑ cleaved caspase-3 ↑ Bax/Bcl-2 ratio cytochrome c release ↑ TUNEL+ nuclei	↓ cell survival ↑ apoptotic cell death	WB, qPCR, TUNEL	Rat genioglossus (CIH; HFD) [138,145,159]

-: no change; n.d.: not determined; ↑ increase; ↓ decrease; BNIP3: BCL2/adenovirus E1B 19 kDa protein-interacting protein 3; Bax: BCL2-associated X protein; Bcl-2: B-cell lymphoma 2; Cyt-c: cytochrome c; DRP1: dynamin-related protein 1; EM: electron microscopy; ERRα: estrogen-related receptor alpha; HFD: high-fat diet; HIF-1α: hypoxia-inducible factor 1 alpha; LC3: microtubule-associated protein 1 light chain 3; Mfn2: mitofusin 2; miR-17-5p: microRNA-17-5p; mtDNA: mitochondrial DNA; NRF1: nuclear respiratory factor 1; PGC-1α/PGC-1β: peroxisome proliferator-activated receptor gamma coactivator-1 alpha/beta; qPCR: quantitative polymerase chain reaction; SIRT1: sirtuin 1; TUNEL: terminal deoxynucleotidyl transferase dUTP nick end labeling; WB: Western blot.

**Table 7 ijms-26-10562-t007:** Oxidative stress and antioxidant/pro-oxidant effects in experimental models.

Oxidative Stress/Findings	Detection/Markers	Functional Evaluation	Antioxidant/Pro-Oxidant Effect	Sample/Model
↑ ROS [145]	DHE	↓ expression of mitochondrial function-related genes	n.d.	Rat, genioglossus/CIH
↑ lipid peroxidation ↑ oxidized DNA [138]	4-HNE 8-OHdG	n.d.	n.d.	Rat, genioglossus/HFD
↑ ROS (mitochondrial) [138]	MitoSOX Red	n.d.	n.d.	Rat, C2C12 myoblasts/Palmitic acid
↑ ROS [138,159]	DHE	n.d.	n.d.	Mouse, genioglossus/CIH
↑ lipid peroxidation ↓ antioxidant enzyme [174]	MDA GSH-Px	n.d.	n.d.	Rat, genioglossus/CIH
transcriptional changes associated with the response to ROS [160]	RNAseq	Electromyography	n.d.	Rat, genioglossus/CIH
No changes in protein oxidation [140]	Protein free thiol and carbonyl content			Rat, sternohyoid/CIH
↑ Mitochondrial ROS ↑ lipid peroxidation ↓ antioxidant enzymes [175]	DCFH-DA probe in isolated mitochondria, MDA Activities of superoxide dismutase, catalase and glutathione peroxidase	In vitro electrophysiological study	- Genistein: Decreased mtROS and lipid peroxidation Increased activities of GPx, CAT and SOD Increased fatigue resistance	Rat, genioglossus/CIH
n.d. [176]	n.d.	In vitro electrophysiological study	- BSO (pro-oxidant): ↑ CIH induced fatigue	Rat, sternohyoid/CIH
n.d. [148]	n.d.	In vitro electrophysiological study tested in the presence and absence of the antioxidant	- NAC (antioxidant): no effect - Tempol (superoxide scavenger): improved muscle force and performance	Rat, sternohyoid/CIH
n.d. [177]	n.d.	In vitro electrophysiological study	- Tempol (superoxide scavenger): chronic treatment for 9 days, reversed CIH induced weakness	Rat, sternohyoid/CIH
n.d. [178]	n.d.	In vivo EMG study	- Tempol, administered systemically, increased EMG activity	Zucker rat, genioglossus/obesity

ROS = reactive oxygen species; DHE = dihydroethydium; 4-HNE = 4-hydroxynonenal; 8-OHdG = 8-hydroxy-2′-deoxyguanosine; MDA = malondialdehyde; GSH-Px = glutathione peroxidase; DCFH-DA = 2,7-Dichlorodihydrofluorescein diacetate; EMG = electromyography; CIH = chronic intermittent hypoxia; HFD = high-fat diet; NAC = N-acetylcysteine; BSO = buthionine sulfoximine; n.d. = not determined. ↑ = increased; ↓ = reduced.

**Table 8 ijms-26-10562-t008:** Summary of mitochondrial alterations in experimental models of OSA.

Domain	Main Findings
Bioenergetics	↓ respiratory chain complexes ↓ mitochondrial membrane potential ↓ ANT1
Oxidative stress	↑ ROS and lipid peroxidation ↑ mtROS ↑ lipid peroxidation, DNA oxidation ↓ antioxidant enzymes (SOD, catalase)
Mitophagy/Quality control	↓ mitochondrial biogenesis ↑ fission ↑ mitophagy
Apoptosis	↑ DNA fragmentation (TUNEL) ↑ Bax/Bcl-2 ↑ cytoplasmic Cyt-c and ↓ mitochondrial Cyt-c ↑ cleaved Caspase-9, Caspase-12, Caspase-3

ΔΨm: mitochondrial membrane potential; ANT1: adenine nucleotide translocase 1; ROS: reactive oxygen species; SOD: superoxide dismutase; PGC-1β: peroxisome proliferator-activated receptor-γ coactivator-1β; DRP1: dynamin-related protein 1; Mfn2: mitofusin-2. ↑ increase; ↓ decrease.

## Data Availability

No new data were created or analyzed in this study. Data sharing is not applicable to this article.

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
