# Peer review of "The Role of Mitochondria in Obstructive Sleep Apnea: Implications for the Upper Airway Muscles"

_ijms, 2025, doi:10.3390/ijms262110562_

Round 1
Reviewer 1 Report
Comments and Suggestions for Authors
This is an iteresting review that will contribute significantly to the field. Nevetheless, I would like to suggest to the authors to address the following revisions that may improve your manuscript.
.It is not clear the relation between sections 4 and 5, try to search for references that include the miRNAs or proteins that are altered in OSA that were mentioned in Section 4. Other way, the article look like two separated reviews.
Please check the attached file for detail comments.

Author Response
We sincerely thank you for your thorough review of our manuscript and for your insightful comments, all of which have been carefully considered and incorporated, contributing significantly to the improvement of the text.
Please find our detailed responses to your comments below.
Comment: It is not clear the relation between sections 4 and 5,
Response: We have added a paragraph in the end of section 4 to improve the connection between sections 4 and 5, as follows:
“In OSA, the upper airway muscles play an important role in maintaining the airway open, particularly during repeated cycles of hypoxia and reoxygenation. Thus, mitochondrial integrity and function are crucial for these muscles, yet this aspect has received little attention in previous reviews. To better understand muscle dysfunction and the pathophysiology of OSA, it is essential to clarify how the disease affects mitochondrial function in the upper airway muscles. In the next section, we summarize the current evidence on these alterations and their functional implications.
”
Comment: try to search for references that include the miRNAs or proteins that are altered in OSA that were mentioned in Section 4.
Response: As suggested by the reviewer, we added Tables 4 and 5 in Section 5 showing the proteins and miRNAs that were reported as altered in OSA. In Table 2, we also added miR-17-5p which is related to mitochondrial quality control, targeting Mfn2.
Comment: Other way, the article look like two separated reviews.
Response: In the introduction, we added a text explaining that we have first provided background information summarizing the general concepts to help readers understand the focus of our review.
“This review focused on the mitochondrial involvement in OSA, specifically synthesizing findings on the impact on upper airway muscles. To frame this discussion, we first provided a concise background on OSA, upper airway muscle function, and the role of hypoxia in mitochondrial function, summarizing the general concepts that readers needed to understand the focus of our review.”
Detailed comments:
- Abstract and Introduction
Abstract (Lines 9-20)
Comment: • While the abstract clearly states the focus, it could benefit from a more explicit mention of the novelty or unique angle this review brings compared to existing literature on OSA and mitochondria. For instance, emphasizing the specific focus on upper airway muscles as a less explored area.
Response: We have reformulated the abstract to cover these points.
Introduction (Lines 25-51)
Comment: • The introduction provides a good overview of OSA and the general role of mitochondria. However, the transition from general mitochondrial function to its specific role in upper airway muscles in OSA could be smoother and more emphasized.
Response: We have improved the transition as suggested by the reviewer. The order of the paragraphs has been changed (the former second paragraph is now third, and the former third paragraph is now second). In addition, the second paragraph (now third) has been reformulated as follows:
“Mitochondria are the primary source of cellular energy and important regulators of redox balance. They are central to cellular adaptations to hypoxia but are also particularly susceptible to hypoxia-induced dysfunction. In OSA, repeated cycles of CIH place particular stress on the upper airway muscles, which depend heavily on efficient mitochondrial function to sustain activity. Under hypoxia, mitochondria may undergo changes in the electron transport chain, as well as in their structure and signaling pathways. Consequently, the generation of reactive oxygen species (ROS) can increase and compromise the function of these muscles. “
- Obstructive Sleep Apnea: Clinical and Pathophysiological Overview (Lines 53-117)
Comment: • This section provides a comprehensive overview. However, some prevalence estimates (e.g., "1.2% to 50%") are very broad. While acknowledged as variable, narrowing this range or providing context (e.g., by age group, geographical region, or diagnostic criteria) could enhance precision.
Response: Because prevalence estimates vary widely and are often stratified by factors such as age, sex, disease severity, and comorbidities, we decided to report the prevalence in the general population as presented in the systematic review by Senaratna et al. (2017), as follows.
“In the general adult population, the prevalence of OSA is estimated at 9–38% for mild cases and 6–17% for moderate cases, with higher rates observed in men and with a progressive increase with age [12].”
Comment: • The discussion on "Variable responses to hypoxia" (Lines 97-100) is interesting but feels a bit disconnected from the main flow.
Response: We have reformulated this discussion to improve the flow, as follows:
“The susceptibility to OSA can be modulated by individual differences, environment and lifestyle (smoking, alcohol consumption, and excess body weight). Physiological responses may vary and influence how the disease manifests. In some patients, minor fluctuations in blood gases can trigger exaggerated ventilatory responses [33-35] [11, 36, 37] while in others, mild hypocapnia may suppress respiratory drive and reduce activation of pharyngeal dilator muscles, thereby increasing the risk of airway collapse [1].”
- Upper Airway Muscle Function in OSA: Anatomy and Physiology (Lines 122-162)
Comment: • Figure 1 is helpful but could be enhanced. The current image on the left is a general anatomical diagram. Consider adding a subtle overlay or annotation on the anatomical image to specifically highlight the upper airway muscles being discussed (e.g., genioglossus, palatopharyngeus). This would reinforce the connection between the anatomy and the functional diagrams.
Response: Figure 1 was reformulated by adding all the cited muscles on the anatomical figure.
Comment: • The section sets the stage for mitochondrial involvement (Lines 158-162), but this transition could be more robust. Explicitly state the hypothesis that mitochondrial dysfunction directly contributes to the mechanical failure of these muscles during sleep, setting up the subsequent sections more clearly.
Response: We have added the following text at of section 3 to highlight the important role of mitochondria and improve the transition to the next section.
“To counteract these forces, the upper airway muscles are constantly in demand and heavily rely on mitochondrial energy production to sustain both tonic and phasic contractions, thereby keeping the airway open. Due to this dependence, the recurrent cycles of intermittent hypoxia and reoxygenation in OSA position mitochondria at the center of upper airway muscle performance and stability. Hypoxia, in turn, drives metabolic adaptations that directly involve mitochondrial pathways.”
- Hypoxia and Mitochondria: An Overview (Lines 178-349)
4.1. Metabolic Adaptation (Lines 179-286)
Comment: • This section is very detailed on HIF-1 and its role in metabolic reprogramming. Table 1 is excellent, but ensure all abbreviations in the table are defined in the main text or in the abbreviations list.
Response: We have checked the use of abbreviations throughout the text and in the list.
Comment: For a review focused on muscles, briefly explain why the metabolic shift towards glycolysis (and away from OXPHOS) is particularly detrimental or impactful for muscle function, especially for muscles requiring sustained activity.
Response: We have added a discussion about the metabolic consequences in the muscle with sustained activity in section “5.2 Mitochondrial function, Human studies”, with the following text:
“Biopsies from the upper airway muscles of OSA patients revealed selective metabolic changes. In human uvular samples, glycolytic enzyme activities were more consistently altered, with increased activity of phosphofructokinase (PFK), glyceraldehyde-3-phosphate dehydrogenase (GAPDH), and glycogen phosphorylase (PYG) in patients with OSA compared to snorers [112] [113]. These findings suggest a shift in energy metabolism toward glycolysis, which can be considered an adaptation to hypoxia [53] and is consistent with the energy required during rapid bursts of muscle activity during apnea episodes. However, this shift may be detrimental to upper airway muscles, which function continuously, as glycolysis provides limited ATP and promotes lactate accumulation, leading to acidification and potential muscle damage.”
4.2 Adaptations in Mitochondrial Quality Control (Lines 288-346)
Comment: • Please, clarify the direct link between these general mitochondrial quality control mechanisms and their specific relevance to the upper airway muscles in OSA. While implied, a sentence or two explicitly connecting the general mechanisms to the unique demands or stresses on these muscles would be beneficial.
Response: To clarify the link between mitochondrial quality control and upper airway muscles in OSA, we have added the following text at the end of Section 5.2.
“Under physiological conditions, a balance between mitophagy and biogenesis maintains energy supply and cell survival by removing damaged mitochondria and generating new ones [150]. When this balance is disrupted, through excessive mitophagy or insufficient biogenesis, mitochondrial decline, leading to muscle weakness and impaired ability to maintain upper airway patency.”
4.3. ROS Generation and Signaling (Lines 347-380)
Comment: • Propose potential reasons for this variability beyond methodological differences, such as inter-individual variability in antioxidant capacity, disease severity, or genetic predispositions in OSA patients. This could add depth to the "Future Perspectives" section.
Response: We have expanded the section “ROS generation and signaling” to include additional factors that may influence ROS production in humans (comorbidities, genetic susceptibility, lifestyle/dietary influences).
“In humans, the production of ROS may also be influenced by additional factors beyond hypoxia itself, as well as the detection methods used. The intensity of oxidative stress can also be affected when there is an association of other conditions with increased ROS, such as cardiovascular and metabolic diseases [101]. Other conditions that can additionally affect the susceptibility to oxidative stress are genetic features, such as the presence of polymorphisms in genes related to antioxidant systems [102]; lifestyle or dietary factors [103].”
Related aspects are further addressed in section “5.4. Oxidative stress”, where the roles of disease severity and comorbidities in shaping oxidative stress markers are discussed.
“In a study with a large cohort of 402 individuals, including 71 controls and patients with OSA of varying severity, Peres et al. (2020) conducted a muliple regression anayses for lipid peroxidation (8-isoprostane), DNA oxidation (8-OHdG), and superoxide dismutase (SOD), with adjustments for demographic, clinical, and lifestyle confounders, including BMI, AHI, comorbidities, statin use, and ethnicity [158]. After adjusting for these factors, only 8-isoprostane levels were found to be independently associated with AHI. This study highlights the importance of including confounding factors in biomarker studies [158].”
- Mitochondrial Alterations in OSA: Influence on Upper Airway Muscles (Lines 381-588)
Comment: • Improvement: This is a core section, but the flow between human studies and experimental models can be disjointed. Consider organizing this section with clear subheadings for "Human Studies" and "Experimental Models (CIH, Lepoblob, HFD)" to improve readability. Within each, discuss findings related to structure, function, quality control, and oxidative stress.
Response: To improve readability, we have thoroughly revised this section and organized the content under the subheadings “Human Studies” and “Experimental Models,” as suggested. Within each subsection, the results are presented separately to ensure a clearer flow between findings.
Comment: • Offer a brief discussion on the challenges of human muscle biopsies in OSA research and how this limits comprehensive understanding, reinforcing the need for experimental models.
Response: As suggested, we have added a brief discussion on the challenges of obtaining human muscle biopsies in OSA research, highlighting the ethical and practical limitations of these procedures and their potential impact on already compromised tissues. This addition reinforces the need for experimental models to investigate mitochondrial alterations in OSA.
The text was added in the last paragraph of section 5.4. Oxidative stress – Human studies.
“Upper airway muscle samples can only be obtained during surgical procedures for OSA, which are associated with significant postoperative pain. Even the removal of small fragments carries the risk of further compromising already vulnerable tissues. These limitations hinder the direct study of upper airway muscles in humans, justifying the use of experimental models to investigate mitochondrial alterations in OSA.”
Comment: • Ensure that all elements in Figure 2 are clearly referenced or explained in the preceding text. For instance, "reduced capillary supply" is mentioned in the figure but might warrant a brief elaboration in the text if not already present.
Response: Capillary supply was mentioned in the text, but in the revised figure we have just mentioned the consistent alterations found in muscle biopsies. So capillary supply has been removed from the figure because it was not a consistent finding.
- Conclusions and Future Perspectives (Lines 594-610)
T Comment: he conclusions summarize the key findings effectively.
The future perspectives are a bit brief. Please expand on specific future research directions. For example:
- Translational Research: How can the understanding of mitochondrial dysfunction in upper airway muscles translate into novel diagnostic tools (e.g., non-invasive biomarkers of muscle health) or therapeutic strategies (e.g., targeted mitochondrial therapies, specific exercise regimens for upper airway muscles)?
- Multi-omics Integration: Discuss the potential of integrating metabolomics, proteomics, and transcriptomics with mitochondrial studies to gain a more holistic view.
- Personalized Medicine: How might individual differences in mitochondrial resilience or response to hypoxia influence OSA severity and treatment outcomes?
- Specific Interventions: Beyond general antioxidant strategies, propose specific compounds or lifestyle interventions that could target mitochondrial health in upper airway muscles
Response: The conclusion and future perspectives section has been revised to more effectively summarize the key findings and to expand on specific future research directions. In particular, we now highlight the need for experimental studies to overcome the limitations of human biopsies, validation of systemic biomarkers, clarifying the primary sources of ROS, possible mitochondria-targeted therapies (antioxidants) and multi-omics approaches.
General Improvements
Comment: •Consistency in Terminology: Ensure consistent use of terms throughout the manuscript (e.g., "upper airway muscles" vs. "oropharyngeal muscles").
Response: The manuscript has been carefully revised to ensure consistent terminology.
Reviewer 2 Report
Comments and Suggestions for Authors
The Authors reviewed the role of mitochondrial dysfunction in the pathophysiology of obstructive sleep apnea, with a focus on its impact on upper airway muscle function. They highlighted that while anatomical and ventilatory control factors are established contributors to the disorder, the influence of impaired mitochondrial function remains underexplored. Considering the critical role of mitochondria in sustaining muscle activity and adapting to hypoxic stress, evidence suggests that mitochondrial abnormalities may contribute to upper airway dysfunction in obstructive sleep apnea.
Overall, the topic of this manuscript is relevant to the scientific community and merits consideration for publication. However, in its current form, the manuscript appears preliminary and lacks the polish of a finalized version, resembling more a draft than a completed review.
The Authors should reference and discuss the pertinent reports that are currently missing.
The presentation and critical interpretation of previous studies need improvement, with integration of these findings into the narrative of the current review.
The Authors should enhance figures and tables to provide more informative content for IJMS Readers.
Author Response
Comment: The English could be improved to more clearly express the research.
Response: We thank the reviewer for this observation. The manuscript has been carefully revised to improve clarity, readability, and consistency in English expression.
The Authors reviewed the role of mitochondrial dysfunction in the pathophysiology of obstructive sleep apnea, with a focus on its impact on upper airway muscle function. They highlighted that while anatomical and ventilatory control factors are established
contributors to the disorder, the influence of impaired mitochondrial function remains underexplored. Considering the critical role of mitochondria in sustaining muscle activity and adapting to hypoxic stress, evidence suggests that mitochondrial abnormalities may contribute to upper airway dysfunction in obstructive sleep apnea.
Overall, the topic of this manuscript is relevant to the scientific community and merits consideration for publication.
Comment: However, in its current form, the manuscript appears preliminary and lacks the polish of a finalized version, resembling more a draft than a completed review.
Response: We thank the reviewer for this feedback. The manuscript has been thoroughly revised to improve its organization, readability, and overall coherence. In particular, sections were restructured, additional content was incorporated, and tables and figures were refined for clarity. These changes ensure that the manuscript now presents as a complete and finalized review.
Comment: The Authors should reference and discuss the pertinent reports that are currently missing.
Response: We thank the reviewer for this comment. We have carefully reviewed the literature and added pertinent reports that were missing, integrating them into the appropriate sections of the manuscript. These additions strengthen the discussion and provide a more comprehensive overview of the topic.
Comment: The presentation and critical interpretation of previous studies need improvement, with integration of these findings into the narrative of the current review.
Response: We thank the reviewer for this helpful suggestion. We have revised the manuscript to improve the integration of previous studies into the narrative, ensuring that their findings are not only presented but also critically interpreted in the context of our review.
Comment: The Authors should enhance figures and tables to provide more informative content for IJMS Readers.
Response: We thank the reviewer for this suggestion. In the revised manuscript, we have included four tables to summarize key findings across human and experimental studies, and we have improved the figures.
Reviewer 3 Report
Comments and Suggestions for Authors
In general terms, the manuscript is overly extensive, which makes it difficult for the reader to follow and understand the content. The authors should aim for greater precision and consider the use of schematic figures or diagrams to facilitate interpretation of the article.
Major comments:
-
The authors need to clearly determine whether OSA should be considered a pathology or not.
-
The relationship between the pathology and mitochondrial function should be further detailed.
-
The verb tense of the stated objective must be in the past.
-
Specifically, the objective at the end of section 1 should also be written in the past tense; this should be revised consistently throughout the manuscript.
-
In section 2, it is recommended to include a subsection on “Risk factors for OSA” and another on “Treatments,” to enhance clarity and comprehension.
-
Why did the authors focus solely on the activity of upper airway muscles? Could other muscle groups also be affected?
-
The authors should provide a clearer description of the review process: how articles were identified, selected, and analyzed.
-
Tables should not contain a separate “References” column. Instead, the reference number should be inserted within the corresponding HIF-1 target (or equivalent entry).
-
The text needs careful formatting, as many paragraphs appear consecutively without spacing, which hinders readability.
-
Line 408: a reference is missing after the authors’ names.
-
The conclusion is excessively long and repeats information already presented. It should be condensed into a single paragraph of 2–3 sentences highlighting the main contribution and findings of the study.
-
The authors should include a section on practical applications in order to clarify the relevance and utility of this article.
Author Response
Comment: In general terms, the manuscript is overly extensive, which makes it difficult for the reader to follow and understand the content.
The authors should aim for greater precision and consider the use of schematic figures or diagrams to facilitate interpretation of the article.
Response: We appreciate this comment and have revised the manuscript to improve focus and readability. The text was reorganized with clearer structure, additional content, tables, and improved figures were included to facilitate reader understanding.
In the introduction, we clarified that it first provides general background information to situate the reader before addressing the specific focus of our review.
“This review focused on the mitochondrial involvement in OSA, specifically synthesizing findings on the impact on upper airway muscles. To frame this discussion, we first provided a concise background on OSA, upper airway muscle function and the role of hypoxia and mitochondria, summarizing the general concepts that readers must know to understand the focus of our review. “
Major comments:
Comment: The authors need to clearly determine whether OSA should be considered a pathology or not.
Response: We thank the reviewer for this comment. To avoid ambiguity, we clarified in the Introduction that OSA is a well-recognized but underdiagnosed sleep-related breathing disorder, citing established literature (Jordan et al., Lancet 2014). We also verified that throughout the manuscript the term “pathology” is used only in reference to tissue-level alterations (e.g., muscle pathology), not to OSA itself.
Comment: The relationship between the pathology and mitochondrial function should be further detailed.
Response: We thank the reviewer for this valuable comment. To strengthen the mechanistic link, we added a statement in the Conclusion highlighting that mitochondrial dysfunction not only arises from intermittent hypoxia but also contributes to OSA pathology by exacerbating muscle weakness and airway collapsibility.
“In summary, the available results indicated that mitochondrial dysfunction and oxidative stress compromised upper airway muscle function in OSA. The combination of impaired bioenergetics, increased ROS, and oxidative damage leading to muscle dysfunction may create a vicious cycle that further weakens dilator muscles progressively. Thus, mitochondrial dysfunction appears to be both a consequence of OSA-related hypoxia and a contributor to disease pathophysiology.”
Comment: The verb tense of the stated objective must be in the past.
Specifically, the objective at the end of section 1 should also be written in the past tense; this should be revised consistently throughout the manuscript.
Response: The objective at the end of Section 1 has been revised to the past tense, and verb tense has been checked throughout the manuscript to ensure consistency.
“This review focused on the mitochondrial involvement in OSA, specifically synthesizing findings on the impact on upper airway muscles. To frame this discussion, we first provided a concise background on OSA, upper airway muscle function, and the role of hypoxia in mitochondrial function, summarizing the general concepts that readers needed to understand the focus of our review. ”
Comment: In section 2, it is recommended to include a subsection on “Risk factors for OSA” and another on “Treatments,” to enhance clarity and comprehension.
Response: We thank the reviewer for this suggestion. As our review is focused on mitochondrial alterations in upper airway muscles, we believe that dedicated subsections on risk factors and treatments would shift the emphasis toward clinical aspects that are outside the scope of the article. Nonetheless, to provide sufficient context for readers, risk factors (e.g., obesity, comorbidities, lifestyle factors) and treatments (e.g., CPAP, muscle training, pharmacological strategies) are already addressed in the introduction.
Comment: Why did the authors focus solely on the activity of upper airway muscles? Could other muscle groups also be affected?
Response: We thank the reviewer for this question. Our focus was placed on upper airway muscles because they are the key structures responsible for maintaining airway patency and are directly exposed to recurrent hypoxia and mechanical stress in OSA. While other muscle groups may also be affected, a comprehensive review of systemic muscle involvement would broaden the scope considerably. To clarify this point, we added a sentence at the end of Section 1 explicitly stating that this review focuses on upper airway muscles.
“Although systemic effects on other muscle groups may also occur in OSA, this review focused specifically on upper airway muscles because of their central role in maintaining airway patency and their direct exposure to recurrent hypoxia.”
Comment: The authors should provide a clearer description of the review process: how articles were identified, selected, and analyzed.
Response: We added the information that the review is narrative and described how the article search was performed, as follows:
“This review was narrative in scope. References were identified primarily through PubMed searches using the key terms “mitochondria,” “obstructive sleep apnea,” “muscle,” “sleep,” and “hypoxia,” applied individually or in combination. Additional keywords relevant to specific topics were also used to broaden the discussion. For the specific theme of mitochondrial involvement in upper airway muscles in OSA, we aimed to include all available original studies in both patients and experimental models. Background information was summarized selectively to provide context.”
Comment: Tables should not contain a separate “References” column. Instead, the reference number should be inserted within the corresponding HIF-1 target (or equivalent entry).
Response: The tables have been reformatted accordingly, with reference numbers now inserted within the corresponding entries instead of in a separate “References” column.
Comment: The text needs careful formatting, as many paragraphs appear consecutively without spacing, which hinders readability.
Response: The manuscript has been carefully reformatted to ensure appropriate paragraph spacing and improved readability throughout.
Comment: Line 408: a reference is missing after the authors’ names.
Response: We have added the reference number to Vuono et al.
Comment: The conclusion is excessively long and repeats information already presented. It should be condensed into a single paragraph of 2–3 sentences highlighting the main contribution and findings of the study.
Response: The conclusion and future perspectives section has been revised to more effectively summarize the key findings and to expand on specific future research directions.
Comment: The authors should include a section on practical applications in order to clarify the relevance and utility of this article.
Response: We thank the reviewer for this valuable suggestion. While we felt that a separate section would extend the manuscript beyond its intended scope, we have incorporated the relevant content into the “future perspectives”, to ensure the topic is addressed.
The main practical application that could result from increasing the understanding on this subject is the potential of targeted-mitochondria antioxidant therapy to improve the upper airway muscle strength.
Round 2
Reviewer 2 Report
Comments and Suggestions for Authors
The Authors reviewed current evidence on the role of mitochondrial dysfunction in OSA, a prevalent yet under-diagnosed condition characterized by recurrent upper airway obstruction and intermittent hypoxia. The review elegantly highlights that while anatomical and ventilatory control factors are established contributors to OSA, the contribution of mitochondrial abnormalities in upper airway muscle function remains insufficiently understood.
Major concerns:
-The Authors need to provide a more systematic evaluation of the available evidence, specifying inclusion criteria for the studies discussed.
-They should elaborate on the mechanisms by which mitochondrial dysfunction affects upper airway muscle performance during sleep;
-The Authors must clarify whether mitochondrial alterations are primary contributors or secondary adaptations to intermittent hypoxia.
-Data should be summarized regarding mitochondrial bioenergetics, oxidative stress, and apoptosis in upper airway muscles in OSA models;
-The Authors should discuss whether mitochondrial-targeted therapies have been evaluated in preclinical or clinical settings of OSA.
-They must integrate evidence linking mitochondrial dysfunction with other systemic consequences of OSA, such as cardiovascular and metabolic disturbances.
-The Authors should provide a graphical summary illustrating proposed mitochondrial mechanisms in OSA pathophysiology.
-They should better highlight key research gaps and methodological limitations in the current literature.
Author Response
We sincerely thank the reviewer for the careful evaluation and constructive comments that helped to improve our manuscritp. We have thoroughly revised the paper and addressed each comment individually, as explained below.
Comment: The English could be improved to more clearly express the research.
Response: The text was extensively revised to improve clarity and readability.
Comment: The Authors need to provide a more systematic evaluation of the available evidence, specifying inclusion criteria for the studies discussed.
Response:
As suggested, we have provided a systematic literature search which was described in a new section ( “Literature search and study selection”) and a supplementary table (Table S1), containing the search strings and number of retrieved records.
Here is the description added to the text:
- Literature search and study selection
A PubMed search was conducted to identify studies investigating mitochondrial aspects in the upper airway muscles in patients with OSA or experimental models.
The inclusion criteria were: (a) studies examining the upper airway muscles or comparable muscle cells with results related to muscle and mitochondrial structure, mitochondrial function, mitochondrial quality control or oxidative stress in OSA patients or OSA-related experimental models, (b) original research articles, and (c) written in English. Reviews, case reports, and studies not directly related to the thematic scope of this review were excluded. The search had no date restriction but was last updated on 13 October 2025.
We examined studies that assessed mitochondrial structure, function, or regulatory pathways (including biogenesis and mitophagy), together with morphological and functional features of upper airway muscles when related to mitochondrial alterations or oxidative stress.
Due to the scarce number of studies focusing specifically on oxidative stress in upper airway muscles in humans, we also included clinical studies evaluating systemic oxidative stress markers (in blood or exhaled breath) to provide a broader insight into the oxidative status in patients. The selected studies met all the following criteria: (1) quantification of specific molecular oxidative biomarkers; (2) assessment of the association with OSA severity or its therapeutic modulation; (3) measurement performed in blood or other non-invasive biological fluids; and (4) primary biochemical investigation of oxidative stress.
The search strategy was structured into four thematic categories: (1) muscle and mitochondrial structure, (2) mitochondrial function, (3) mitochondrial quality control, and (4) oxidative stress. For each category, search strings were constructed using multiple conceptual blocks that combined disease-related, anatomical, muscle-related, mechanistic and molecular terms, and species-specific identifiers. Boolean operators (AND/OR) and exclusion filters (“NOT”) were applied to refine results, and additional limitations such as language and species were set using PubMed filters.
Retrieved records were screened through the titles, abstracts and text contents to identify studies relevant to one or more of the thematic categories. Articles not addressing these aspects or focusing on unrelated tissues or outcomes were excluded. The full PubMed search strings and the number of records retrieved for each category are presented in Supplementary Table S1.
Comment: They should elaborate on the mechanisms by which mitochondrial dysfunction affects upper airway muscle performance during sleep;
Response:
We added this discussion in the “conclusions and future perspectives” section.
Lines 935-956
At first, hypoxia may induce adaptive changes to maintain muscle function under reduced oxygen availability. Muscle fibers are remodeled, energy metabolism is adjusted, and mitochondrial quality control processes are activated to help maintain sufficient ATP production. However, these compensatory responses have limited capacity, and prolonged exposure to CIH with decreased respiratory chain activity, can lead to increased ROS generation, disrupting the balance between ROS and antioxidant defenses. Oxidative damage may affect mitochondria, especially the OXPHOS system, leading to mitochondrial dysfunction and further predisposing to increased ROS production. This way, a self-perpetuating cycle of mitochondrial dysfunction and oxidative stress is established, compromising upper airway muscle function, increasing airway collapsibility, and further exacerbating hypoxic events. A similar mechanism was previously proposed, considering that adaptation to hypoxia starts with a compensatory hyperactivation of the upper airway dilator muscles, but later a structural injury is established, leading to reduced effectiveness of these muscles and hypoxia, which again promotes hyperactivation of the dilator muscles [122]. Muscles with insufficient OXPHOS are unable to maintain full function, and as oxidative lesions increase, the tendency is for a progressive decline in muscle strength and damage. Thus, what begins as an adaptive response to hypoxia may progress into a maladaptive state culminating with structural injury.
Comment: The Authors must clarify whether mitochondrial alterations are primary contributors or secondary adaptations to intermittent hypoxia.
Response: Mitochondrial alterations are initially part of the adaptations to intermittent hypoxia, however, with time the protective mechanisms for mitochondrial alterations (quality control) and for oxidative damage are insufficient and mitochondrial deficiency and increase ROS generation act as contributors to the muscle weakness.
This is explained in lines 935-956
Thus, what begins as an adaptive response to hypoxia may progress into a maladaptive state culminating with structural injury.
Comment: Data should be summarized regarding mitochondrial bioenergetics, oxidative stress, and apoptosis in upper airway muscles in OSA models;
Response:
Ok. Table 8 with a summary of the results in experimental models was added.
Comment: The Authors should discuss whether mitochondrial-targeted therapies have been evaluated in preclinical or clinical settings of OSA.
Response:
Mitochondrial-target therapies have not been evalutated in preclinical or clinical studies in OSA models or patients. This information was added the section Conclusions and Future Perspectives.
Lines 980-83
Although MitoQ has undergone clinical evaluation in other diseases, no preclinical studies or registered clinical trials have yet tested MitoQ or MitoTEMPO in OSA (clinicaltrials.gov, search on 10/23/2025).
Comment: They must integrate evidence linking mitochondrial dysfunction with other systemic consequences of OSA, such as cardiovascular and metabolic disturbances.
Response:
Similar results with mitochondrial dysfunction, increased ROS and mitochondrial fragmentation have been described in endothelial cells, pancreatic islets and myocardium. Mito-targeted antioxidants were able to restore endothelial and myocardium function.
Lines: 957-969
Similar to the upper airway muscles, mitochondrial alterations have also been reported in diseases frequently associated with OSA, such as cardiovascular and metabolic disorders [179]. CIH caused ultrastructural changes in mitochondrial morphology, reduced SOD activity, increased mtROS production, and induction of apoptosis in endothelial cells [180]. Pancreatic islets exposed to CIH displayed increased mtROS and impaired insulin synthesis [181]. In the myocardium, in addition to mitochondrial ultrastructural disruption, impaired cardiac remodeling and dysfunction were observed, associated with increased mitochondrial fragmentation, reduced oxygen consumption, and increased ROS [182, 183]. Treatment with mitochondrial-targeted antioxidants, such as MitoTempo or MitoTempol, reversed endothelial apoptosis and restored insulin synthesis and glucose homeostasis in pancreatic islets [180, 181].
Comment: The Authors should provide a graphical summary illustrating proposed mitochondrial mechanisms in OSA pathophysiology.
Response:
The graphical summary was provided as suggested.
Comment: They should better highlight key research gaps and methodological limitations in the current literature.
Response: These points were highlighted in the Conclusions and Future Perspectives section. We pointed for the confirmation of mitochondrial ROS as the main source of ROS in chronic intermittent hypoxia, time window for reversibillity of muscle alterations, search for a reliable biomarker for mitochondrial alterations and oxidative stress in blood samples.
Lines: 983-1007
It would also be important to confirm that mtROS are the major contributors to the CIH effects in muscle, as this has been little explored by previous studies, and would support the rationale for mitochondria-targeted therapeutic strategies.
Given the relevance of treatment to improve upper airway muscle function, an important question is the reversibility of CIH-induced muscle abnormalities. Some of the CIH-induced changes, such as fiber-type remodeling (increased proportion of type II fibers), appear to be reversible. This reversibility is suggested by clinical studies using CPAP, which demonstrate the recovery of muscle structure after treatment, and by in vitro experiments showing restoration of muscle performance after antioxidant treatments. However, myopathic features, reflecting muscle damage, were observed in the upper airway muscles in some patients and in the dog model, suggesting that irreversible damage may occur with prolonged disease duration. Therefore, it is essential to determine the time window during which these alterations remain reversible.
Another challenge in clinical trials is the lack of a reliable marker for mitochondrial alterations and oxidative stress, due to difficulties in directly evaluating the upper airway muscles. At this point, experimental studies will be particularly helpful in finding these answers. The use of multi-omics approaches and longitudinal study designs may help clarify the temporal progression from adaptive to maladaptive mitochondrial responses and identify the timing of therapeutic intervention for greatest effectiveness. Future research should also identify reliable indicators of oxidative stress or mitochondrial function for patient monitoring and clinical trials.
Reviewer 3 Report
Comments and Suggestions for Authors
The authors have made a great effort to improve the text and have responded satisfactorily to my comments.
Author Response
We thank the reviewer comment. The reviewer did not have any other suggestions or comments to be addressed. All comments were addressed accordingly in the previous review.
Round 3
Reviewer 2 Report
Comments and Suggestions for Authors
-